# Research on the Evaluation and Spatial Characteristics of China’s Provincial Socioeconomic Development and Pollution Control Based on the Lotka–Volterra Model

**DOI:** 10.3390/ijerph20054561

**Published:** 2023-03-04

**Authors:** Xue Zhou, Jiapeng Wang

**Affiliations:** School of Economics, Qingdao University, Qingdao 266071, China

**Keywords:** industrial pollution, domestic pollution, socioeconomic development, Lotka–Volterra model, spatial autocorrelation, spatial heterogeneity

## Abstract

Aims: To evaluate the degree of mutualism between socioeconomic development and industrial and domestic pollution in provinces of China and to analyze the differences in spatial characteristics between their regions. Methods: This study used the HDI to measure socioeconomic development and the Lotka–Volterra model to group and estimate the force-on and mutualism degree indexes of industrial and domestic pollution and socioeconomic development in 31 provinces of China, which were then used to them. Then, the study calculated the global and local Moran’s *I* under different space weights matrices to analyze their spatial autocorrelation and heterogeneity. Results: The research showed that in 2016–2020, compared with 2011–2015, the number of provinces where socioeconomic development and industrial pollution control mutually promoted each other was approximately the same, while the number of provinces that promoted each other’s effectiveness with domestic pollution control was reduced. There were many provinces with industrial pollution ranked in the S-level, while most provinces placed a different emphasis on industrial and domestic pollution control. The rank in China tended to be spatially balanced in 2016–2020. There was a negative spatial autocorrelation between the ranks of most provinces and neighboring provinces in 2011–2020. The ranks of some eastern provinces showed a phenomenon of a high–high agglomeration, while the ranks of provinces in the western region were dominated by a high–low agglomeration.

## 1. Introduction

China has shifted to a high-quality development stage, with innovation as the primary driving force, coordination as the endogenous feature, and green as the universal form. Following the concept of green development and correctly grasping the relationship between ecological environmental protection and socioeconomic development, there are inherent requirements for promoting high-quality development. Taking coordinated regional development as the goal and paying attention to the solution of unfair and unbalanced problems between regions is an inevitable requirement for high-quality development to adapt to the changes in the main contradictions in Chinese society. However, rapid economic development is accompanied by ecological damage and environmental pollution. *The Bulletin on the Second National Census of Pollution Sources* shows [1] that pollution is at a high level, and the emissions of pollutants from some domestic sources, such as chemical oxygen demand and ammonia nitrogen, have exceeded those of industrial sources. Protecting and improving the ecological environment has become an important task for socioeconomic development. The 14th Five-Year Plan [2] clearly states that pollution prevention and control actions should continue to be carried out, in addition to strengthening the coordinated control of multiple pollutants and regional coordinated governance, as well as promoting the comprehensive green transformation of socioeconomic development. The coordinated prevention and control of industrial and domestic pollution, coordinated ecological protection, and development of the social economy will provide a strong impetus for higher quality regional linkages. This paper attempts to explore the relationship between industrial and domestic pollution prevention and socioeconomic development in China, the degree of coordination between the two, and its spatial distribution characteristics. By using the Lotka–Volterra model to rate the coordination relationship, the spatial characteristics of interregional ratings are explored by measuring the global and local Moran indexes, to provide a reference for the policy formulation of the Chinese government’s socioeconomic development, coordinated development of pollution prevention and control, and coordinated regional development.

In recent years, the analysis of the relationship between pollution and development has mainly focused on the relationship between air pollution, water pollution and economic development. Miah et al. [3] analyzed industrial enterprises, such as sugar factories and distilleries, that produce a lot of wastewater. They believed that industrial development is an important cause of river water pollution in Bangladesh. Sakti et al. [4] used CO, NO_2_ and SO_2_ data to analyze the relationship between air pollution and human activities in Southeast Asia and characterize the distribution of these air pollutants. Mele et al. [5] believed that there was a significant correlation between India’s economic growth and pollution, in which pollution was measured by CO_2_ and NOx. Halkos et al. [6] studied data from 119 countries and found that the EKC hypothesis of economic growth and CO_2_ emissions was confirmed in high- and upper-middle-income countries, and that there was a two-way Granger causal relationship between GDP per capita and CO_2_ per capita in all countries. At present, research on the relationship between pollution and economic development in China is mainly divided into two aspects. On the one hand, the impact of smog or water pollution caused by industrial pollution on economic development is studied. Zhao et al. [7] used China’s provincial PM_2.5_ from 2004 to 2016 as the pollution index, arguing that smog pollution will significantly reduce the quality of China’s economic development. Cheng et al. [8] conducted a spatial analysis of urban water pollution in Wuhu and concluded that domestic pollution, agricultural pollution and urban-surface-runoff pollution contributed to the main pollution load. Fang et al. [9], based on China’s PM_2.5_ emissions data from 2000 to 2014, concluded that industrial development was the main reason for the increase in emissions. On the other hand, the EKC curve of pollution and economic development is also analyzed. Jiang et al. [10] verified the existence of the EKC by analyzing the “2 + 26” urban air pollution and economic growth, and determining the U-shaped relationship between the two. Liu et al. [11] verified the EKC relationship between industrial wastewater and economic growth in the Nansi Lake region of China.

Some scholars have analyzed the relationship between pollution and economic development from a spatial perspective. Li et al. [12] used a spatial Doberman model analysis of major Chinese cities from 2000 to 2012 and concluded that industrial agglomeration has obvious spatial spillover effects on haze pollution. Liang et al. [13] analyzed the relationship between environmental pollution and economic development, industrial structure, regulatory measures and infrastructure construction using air pollution and water pollution as the main pollutants. Dang et al. [14] analyzed the impact of imports and exports as part of economic development on pollution, and found that the domestic wastewater pollutants caused by exports decreased, while solid waste increased partially. Liu et al. [15] used Shandong, China as the research area and argued that there was no coordination between industrial and domestic water pollution and socioeconomic development. Some scholars have studied the spatial characteristics of pollution and development, such as Gong et al. [16], who found that China’s environmental pollution has a positive spatial correlation by calculating the global and local Moran index and concluded that industrial structure and population density are the main variables affecting real estate and environmental pollution through a spatial mediation model. Cheng et al. [17] used a dynamic spatial panel model to analyze the association between economic development and smog pollution in 285 cities in China. The analysis of pollution focuses on the relationship between pollution sources such as water, solid waste or air pollution and industrial and economic development. The selection of pollutants is not comprehensive, and the development of human society does not only involve industrial or economic development. Mainstream research does not consider the impact of human society on pollution.

At present, with the changes in the main contradictions in China today, the importance of analyzing socioeconomic development has gradually emerged to match the needs of the people for a better life. Compared with studies that only analyze economic development, analyzing socioeconomic development is more comprehensive and relevant. At present, the international mainstream focuses on the socioeconomic development represented by health and analyzes the relationship between health and pollution. Velis et al. [18] analyzed the relationship between the Social Progress Index and Corruption Perceptions Index and other poorly managed municipal solid waste. Trifkovic et al. [19], assuming that population agglomeration is a trend, argued that urban areas need to develop in a way that reduces health risks to maintain their existing advantages. Wang et al. [20] classified wastewater as industrial, domestic and agricultural wastewater, and they found that people would think that an increase in industrial and domestic wastewater would affect their health. Liang et al. [21] confirmed the EKC effect between pollution and urbanization and economic development.

Socioeconomic development and pollution have rarely been comprehensively studied in recent Chinese academic circles. Meng et al. [22] analyzed the relationship between environmental pollution and socioeconomic development in the Beijing–Tianjin–Hebei region by measuring the socioeconomic development of the resident population, the income and gross product of different industries, and pollution by investment in industrial pollution, energy consumption and industrial waste gas. Yang et al. [23] analyzed the causes of urban air pollution in Northeast China using population, GDP, green space area and ratio of different industries as socioeconomic development indicators, and AQI and others as air pollution indicators. Wang et al. [24] analyzed the impact of soil pollutants on socioeconomic development in the Pearl River Delta of China. These studies covered socioeconomic development but the selection of indicators for both sides was not comprehensive. The HDI is a globally recognized indicator of human socioeconomic development. Zhang et al. [25] pointed out that the HDI is a widely used indicator to measure sustainable development as well as socioeconomic development. Zuo et al. [26] proposed the concept of complex indicators, including the HDI, which can be considered from multiple dimensions to assess the progress of a region on complex issues. In addition, some scholars have conducted spatial research on the HDI. Luo et al. [27] constructed the HDI with spatial factors, arguing that it has a spatial spillover effect and that its spatial correlation is increasing year by year.

In summary, although many scholars pay attention to the interrelationship between socioeconomic development and pollution, most of these studies take per capita GDP and industrial pollution as the main indicators, and there are few comprehensive studies on socioeconomic development and different pollution sources. At the same time, there are few relevant studies considering the spatial characteristics of development and pollution coordination, and most of the research in this area has focused on analyzing the spatial relationship between development and pollution. We believe that socioeconomic development, measured using the HDI, should be used to represent development, dividing pollution into two aspects, industrial and domestic pollution, and deeply exploring the degree of coordination between development and pollution and how they are interrelated between regions. Therefore, this study uses the Lotka–Volterra model to rate and analyze the coordination between industrial and domestic pollution and socioeconomic development in 31 provinces directly under the central government in China from 2011 to 2020, and explores the spatial autocorrelation of interregional ranks by measuring global and local Moran’s *I*. This study will help innovate and improve the coordination mechanism of regional socioeconomic development and pollution prevention, and control coordination mechanisms and new mechanisms of regional linkage development. This study will provide reasonable suggestions for government departments to explore new models of the interregional industrial division of labor and cooperation and environmental collaborative governance in order to balance regional differences and achieve a higher quality regional coordinated development pattern.

The structure of the article is as follows: Section 2 explains the data source and introduces the methods and models used. Section 3 explains the results of the indexes and rankings, then analyzes the spatial characteristics of the ranking. Section 4 describes the conclusions and policy recommendations.

## 2. Materials and Methods

### 2.1. Data Sources

This study analyzes the relationship between industrial and domestic pollution and socioeconomic development in 31 provinces. The indicators used, such as average life expectancy and education level involved in the HDI calculation process were obtained from the China Statistical Yearbook. The indicators of chemical oxygen demand and ammonia nitrogen emissions in the industrial and domestic pollution effectiveness indicators were obtained from the China Statistical Yearbook on Environment.

### 2.2. The Proportion of HDI

#### 2.2.1. Provincial HDI Calculation Method

The HDI published by the UNDP encompasses the three dimensions of health, education and income that characterize human welfare. It is accepted by scholars and governments as an important indicator of socioeconomic development. Continuous HDI data for provinces from 2011 to 2020 were not directly available. This study used the HDI formula published by the UNDP in the technical notes in the Human Development Report 2020 [28] and combined the provincial HDI data conversion method in the China National Human Development Report Special Edition [29] to calculate the HDI for 31 provinces from 2011 to 2020. There were no direct data on average life expectancy in non-census years, and the average life expectancy values of provinces in the missing years were completed using the linear method. According to the National Bureau of Statistics, in 2017 the PPP was 4.184. The HDI was calculated with 2011 as the base period as follows: *HDI* = (*I*_*Health*_ × *I*_*Education*_ × *I*_*Income*_)^1/3^(1)
where *I_Health_* is the health index, using the average life expectancy calculation; *I_Education_* is the education index, using the proportion of population with primary school, junior high school, high school and college degree or above; *I_Income_* is the income index, using the per capita gross national income calculated according to the 2017 PPP converted value [30].

#### 2.2.2. Calculate the Proportion of HDI

The HDI of most provinces showed an upward trend between 2011 and 2020, and the change in the proportion of each province in the country indicated the structural change of each province in the target year. This study used the ratio of HDI in a province to the total HDI of the whole country (hereafter referred to as the proportion of HDI) to represent the HDI index of a province to indicate the relative speed of HDI improvement in each province.

Figure 1 shows that the proportion of HDI in Guizhou, Yunnan, Xizang, Shaanxi and Qinghai increased in the target year range. The HDI increase in these five provinces was greater than that of other provinces, among which Guizhou and Yunnan paid more attention to health from 2011 to 2020, and Xizang attached great importance to education, which was the reason for the rapid growth of HDI in these three provinces. The proportion of HDI in Tianjin, Hebei, Zhejiang, Anhui, Shandong, Henan and Xinjiang decreased to varying degrees. The HDI of these provinces did increase, but the growth rate was slower than that of the other provinces. Among them, the five provinces of Tianjin, Hebei, Zhejiang, Shandong and Xinjiang, which had a large decline, encountered bottlenecks in economic development, and the per capita GDP after the deflation of the GDP index fell in the target year range. The proportion of HDI in most other provinces, such as Beijing and Shanxi, did not change substantially in the target year range and was in a state of steady development.

### 2.3. Indicators of Industrial and Domestic Pollution Effectiveness

#### 2.3.1. Calculation Method of Industrial and Domestic Pollution Effectiveness Indicators

In the selection of pollution indicators, the reasons for using aggregate indicators rather than per capita indicators are as follows. First, when the total amount of pollution remains unchanged, the per capita indicator will change with the change in population. A decrease in the per capita pollution index does not mean that the pollution situation will improve when the population increases. Second, some pollution is aggregated, and the negative impact on people does not decrease with the increase in the population in the area.

Based on the above analysis, six pollution indicators were selected for industrial pollution: industrial chemical oxygen demand, industrial ammonia nitrogen emissions, industrial sulfur dioxide emissions, industrial nitrogen oxide emissions, industrial smoke dust emissions and general industrial solid waste generation. Correspondingly, in terms of domestic pollution, six pollution indicators were selected in this study: domestic chemical oxygen demand, domestic ammonia nitrogen emissions, domestic sulfur dioxide emissions, domestic nitrogen oxide emissions, domestic smoke dust emissions and domestic garbage removal and transportation. According to the analysis of Yang et al. [31], passenger transportation contributes approximately 22% to air pollution, freight traffic contributes approximately 17% and motor vehicle exhaust emissions in domestic pollution cannot be ignored. Since motor vehicles are not taken into account in some domestic exhaust gas statistics, this study added motor vehicle emissions to the data of domestic NO_x_ emissions and soot emissions instead of the original data.

Due to the large gap in pollution emissions between provinces in China, it was necessary to take the logarithm of pollution data during standardization to alleviate the evaluation error caused by poor data. The pollution index is a negative index, and the larger the index value is, the more serious the harm of pollution to the economy and society. Thus, this study used a standardized calculation method for negative indicators. The formula is as follows (the original source we refer to is in Appendix A):*x_ij_*′ = 0.6 + {*ln*[*max*(*x_j_*)] − *ln*(*x_ij_*)}/{*ln*[*max*(*x_j_*)] − *ln*[*min*(*x_j_*)]} ∗ 0.4(2)
where *i* represents the year, *j* represents the pollution index, *x_ij_*′ represents the standardized assignment, *ln*[*max*(*x_j_*)] represents the logarithm of the maximum value of the *j* pollution index, and *ln*[*min*(*x_j_*)] represents the logarithm of the minimum value of the *j* pollution index. Other formulas are the same as in the quotation and will not be repeated. Among them, the single type of pollution effectiveness index is the product of the standardized proportion and weight of the pollution index, and the industrial and domestic pollution effectiveness index is calculated by weighting various pollution indicators. The higher the effectiveness index value is, the lower the total amount of pollution. The comprehensive pollution effectiveness index can measure pollution discharge and prevention more comprehensively than a single pollution index.

#### 2.3.2. Calculation of the Effectiveness Indicators of Industrial and Domestic Pollution

Figure 2 and Figure 3 show that Beijing, Tianjin, Shanghai, Henan, Hunan, Shaanxi, Gansu and Qinghai have significantly improved the industrial pollution effectiveness index in the target year range, with the Beijing–Tianjin–Hebei Economic Belt and Shanghai being more developed and having higher pollution prevention and control standards. The industrial pollution management and control of environmental protection departments in these regions achieved remarkable results from 2011 to 2020. The industrial pollution effectiveness index of Nei Mongol, Jiangsu, Anhui, Fujian, Jiangxi, Shandong, Guangdong, Sichuan, Guizhou, Xizang and Xinjiang decreased. With Guizhou and Xizang going through a period of rapid socioeconomic development, pollution emissions were increasing rapidly. Pollution prevention and control effects were relatively insignificant, and the increase in pollution cannot be ignored. In terms of the domestic pollution effectiveness index, Beijing, Tianjin, Shanxi, Nei Mongol, Liaoning, Jilin, Heilongjiang, Shanghai and Gansu improved. Among the several regions with large increases, Beijing and Shanghai had a higher level of socioeconomic development. The government can better practice the concept of environmental protection, and the degree of pollution prevention and control in both categories exceeded the national average. Zhejiang, Fujian, Jiangxi, Hubei, Hunan, Guangdong, Guangxi, Sichuan, Yunnan and Xizang reduced their domestic pollution effectiveness index in the target year range. Sichuan and Xizang increased their industrial and domestic pollution emissions and their socioeconomic development. It is necessary to appropriately increase the intensity of pollution prevention and control. Nei Mongol and Heilongjiang increased their industrial pollution effectiveness index, while their industrial pollution effectiveness index decreased. These two provinces should pay attention to industrial pollution prevention and control.

### 2.4. Model Building

The Lotka–Volterra model is an extension of the logistic model proposed by Lotka and Volterra, which was originally used to model competition between species. Zhang et al. [32] used the Lotka–Volterra model and complex network theory to analyze the synergistic relationship between high-quality industrial development and ecological environmental protection in Shaanxi Province. Zhang et al. [33] used the gray Lotka–Volterra model to study the competition and cooperation between environmental quality, industrial development quality, resource consumption and technological innovation in Shaanxi Province from 2005 to 2019.

The development of a human society depends on the natural environment because the two are in a system. Pollution and socioeconomic development are closely related and the level of coordination between the two profoundly affects the development of the system. Thus, the Lotka-Volterra model can be applied for system analysis. Referring to Chen et al. [34] and related studies to evaluate ecological civilization, a force and symbiosis index between industrial and domestic pollution and the HDI was constructed based on the traditional Lotka–Volterra model competition coefficient. The force index is a one-way index that reflects the effect of pollution on the HDI or the HDI on pollution. The symbiosis index is calculated according to the force index, which indicates the resultant force of the interaction between pollution and the HDI. The higher the value, the more the system tends to coordinate the symbiotic state. The above indicators were used to establish a rank system to determine the degree of coordination and symbiosis between socioeconomic development and pollution in various provinces to evaluate the strength and effectiveness of pollution prevention and control under different levels of socioeconomic development.

In this study, data for provinces from 2011 to 2020 were divided into two groups: 2011–2015 and 2016–2020. The grouping was as follows: first, the data could be grouped according to the time of the 12th and 13th Five-Year Plans, which could reduce the impact of policy factors to a certain extent. Second, since 2016, the accounting methods of some pollution indicators have been adjusted, and the comparison between groups can reduce the evaluation error caused by the change in statistical calibration.

#### 2.4.1. The Lotka–Volterra Model

Taking industrial pollution as an example, the Lotka–Volterra model of the pollution effectiveness index and HDI ratio was constructed, and the model construction process for domestic pollution was similar.

Referring to the research of Guo et al. [35] and related studies, the Lotka–Volterra model for industrial pollution is generally expressed as follows:*dI*(*t*)/*dt* = *R_I_*(*t*)*I*(*t*) × {1 − [*I*(*t*)/*K_I_*(*t*)] + *α*(*t*)[*H*(*t*)/*K_I_*(*t*)]}*dH*(*t*)/*dt* = *R_H_*(*t*)*H*(*t*) × {1 − [*H*(*t*)/*K_H_*(*t*)] + *β*(*t*)[*I*(*t*)/*K_H_*(*t*)]}(3)
where *I*(*t*) indicates the level of industrial pollution in the system and *H*(*t*) indicates the proportion of HDI in the system. *R_I_*(*t*) represents the growth rate of industrial pollution in the system, *R_H_*(*t*) indicates the growth rate of the HDI proportion in the system, *K_I_*(*t*) indicates the maximum possible value of industrial pollution in the system, *K_H_*(*t*) indicates the maximum possible value of the HDI proportion in the system, *α*(*t*) indicates the competition coefficient of the HDI proportion to the industrial pollution effectiveness index and *β*(*t*) indicates the competition coefficient of the industrial pollution effectiveness index to the HDI proportion.

Discretization of Lotka–Volterra model expressions of industrial pollution as Equations (4)–(9) refers to the gray estimation methods of Wu et al. [36] and related studies. For the convenience of discussion, we first changed Equation (3) to the general form:*dI*(*t*)/*dt* = *a_I_I*(*t*) + *b_I_I*(*t*)^2^ + *c_I_I*(*t*) × *H*(*t*)*dH*(*t*)/*dt* = *a_H_H*(*t*) + *b_H_H*(*t*)^2^ + *c_H_H*(*t*) × *I*(*t*)(4)

Taking the upper equation of Equation (4) as an example, the discretization Equation (5) of industrial pollution is obtained, and the data of different time points *t* are brought into Equation (5) to obtain the matrix equation of industrial pollution (7):*I*(*t* + 1) − *I*(*t*) = *a_I_m* + *b_I_m*^2^ + *c_I_m* × *n*(5)
where
*m* = [*I*(*t*) + *I*(*t* + 1)]/2, *n* = [*H*(*t*) + *H*(*t* + 1)]/2(6)
*Y_IN_* = *B_I_a_I_*(7)

The parameter of Equation (4) is estimated as
*A*_*I*_ = [*a*_*I*_, *b*_*I*_, *c*_*I*_]^*T*^ = (*B*_*I*_^*T*^*B*_*I*_)^−1^*B*_*I*_^*T*^*Y*_*IN*_(8)

The coefficients in Equation (3) are obtained according to the matrix
*α*(*t*) = −*c_I_*/*b_I_*, *K_I_*(*t*) = −*a_I_*/*b_I_*, *β*(*t*) = −*b_H_*/*c_H_*, *K_H_*(*t*) = −*a_H_*/*c_H_*(9)

#### 2.4.2. Modeling the Threshold Value Function

Referring to the research of Chen et al. [34], we calculated the force-on-iHDI index as shown in Equations (10)–(12). For that index, *S_I_*(*t*) > 0 indicates that the HDI has a promoting effect on the effectiveness of industrial pollution prevention and control. For the force-on-industrial index in Equation (11), *S_H_*(*t*) > 0 indicates that the effect of industrial pollution prevention and control has a promoting effect on the HDI. The two types of force-on indices were comprehensively analyzed to judge the degree of force coordination. If the force-on index of both directions was greater than 0, the degree of force coordination was considered to be high. The calculation of indices for domestic pollution was similar.
*S_I_*(*t*) = *K_I_*(*t*)/*K_H_*(*t*) + *α*(*t*) = (*a_I_c_H_* − *b_H_a_I_*)/*b_I_a_H_*(10)
*S_H_*(*t*) = *K_H_*(*t*)/*K_I_*(*t*) + *β*(*t*) = (*a_H_b_I_* + *a_H_c_I_*)/*c_H_a_I_*(11)

*S*_1_(*t*) in Formula (12) was calculated based on the force-on index in Equations (10) and (11):*S*_1_(*t*) = [*S_I_*(*t*) + *S_H_*(*t*)]/[*S_I_*^2^(*t*) + *S_H_*^2^(*t*)]^1/2^(12)

This study combined the mutualism degree index with the force-on index to rank the degree of mutualism between HDI and industrial pollution prevention and control. The ranking paid more attention to the impact of the HDI on the effectiveness of industrial pollution prevention and control. Thus, at the higher level of coordination and lower level of coordination symbiosis, the advantages and disadvantages of the HDI industrial stress index are distinguished, where S-level represents the optimal grade and E-level represents the worst level. The rank of domestic pollution prevention and control effectiveness and the HDI was similar to that of industrial pollution in Table 1. The rank of the HDI and industrial or domestic pollution prevention and control effectiveness is hereafter referred to as the industrial or domestic pollution rank.

## 3. Results

### 3.1. Force-On Index

According to Table 2, compared with 2011–2015, the number of provinces that reinforced the HDI and the effectiveness of industrial pollution prevention and control decreased slightly, while the number of provinces that suppressed each other increased slightly. This reflects a slight imbalance between the overall speed of industrial pollution prevention and control and the growth rate of socioeconomic development in the country. The force-on-industrial index in Shandong and Xinjiang turned from negative to positive, and industrial pollution prevention and control began to show results in these two provinces. Between these two sets of years, Shaanxi’s force-on-iHDI index turned from negative to positive, that is, the prevention and control of industrial pollution promoted the improvement of the province’s socioeconomic development level. The force-on indices in Beijing, Liaoning and Qinghai turned from negative to positive, and the coordination degree of the HDI and industrial pollution gradually increased. Among the provinces where the force-on-industrial index turned from positive to negative, the socioeconomic development level of Guizhou and Yunnan improved rapidly, and the effect of industrial pollution prevention and control was relatively slow.

According to Table 3, compared with 2011–2015, the number of provinces where the HDI and the effectiveness of domestic pollution prevention and control were promoted decreased significantly, while the number of provinces that suppressed each other increased slightly. This indicates that the overall domestic pollution prevention and control rate of the country did not match the growth rate of socioeconomic development. At the same time, the force-on indices in Yunnan, Xizang, etc. moved from positive to negative. These areas should control the emission of domestic pollution while increasing the speed of pollution prevention and control by introducing advanced pollution prevention and control technology, improving the level of environmental supervision of government departments, etc., so that it can be coordinated with the growth rate of socioeconomic development. The force-on-domestic index of Zhejiang, Guangdong and Hainan provinces turned from positive to negative, that is, the effectiveness index of domestic pollution prevention and control decreased with the increase in the proportion of HDI. In some developed provinces, such as Zhejiang, Shanghai and Guangdong, the technological progress of domestic pollution prevention and control may be slower than the growth rate of their emissions, resulting in no significant decline in various pollution emissions between 2016 and 2020. Some provinces, such as Hainan, have been approaching zero sulfur dioxide emissions since 2016. The province should continue to maintain the current level of pollution categories with good domestic pollution prevention and control effects while paying attention to the improvement of other types of pollution prevention and control levels. Shaanxi and Qinghai’s force-on-dHDI index turned from negative to positive. The speed of domestic pollution prevention and control in these two provinces was also rapidly increasing, while the level of socioeconomic development was improving.

In summary, the positive and negative changes in the four types of industrial and domestic pollution in provinces were significantly different between the two target year ranges. The degree of coordination between industrial pollution prevention and socioeconomic development in various provinces was significantly better than that of domestic pollution. In recent years, the state has paid more attention to domestic pollution through the implementation of policies such as giving priority to pilot garbage classification in municipalities directly under the central government, provincial capitals and cities with separate plans and replacing buses in built-up areas with new energy vehicles to reduce domestic pollution. It is foreseeable that the degree of coordination of domestic pollution in more provinces will increase in the future.

### 3.2. Mutualism Degree Index and Ranking

The mutualism degree index was calculated based on the force-on index obtained above. This study combined these two types of indices and used the ranking criteria given in Table 1 to analyze the industrial and domestic pollution ranks of 31 provinces from 2011 to 2015 and from 2016 to 2020. The letters above and below each province in Figure 4 and Figure 5 indicate the pollution ranks from 2011 to 2015 and from 2016 to 2020. The S-level represents high-level collaborative symbiosis, the A- and B-levels represent higher-level coordinated symbiosis, the C- and D-levels represent lower-level coordinated symbiosis, and the E-level represents low-level coordinated symbiosis.

As shown in Figure 4, from 2016 to 2020, 14 provinces, such as Beijing and Hebei, were in the stage of advanced mutualism, that is, the S-level. Eleven provinces, such as Shanxi and Nei Mongol, were D-level or E-level, and 8 provinces had a low mutualism degree. The common feature of these provinces was that the effectiveness of industrial pollution prevention and control decreased with the improvement of the HDI. These provinces need to pay more attention to industrial pollution prevention and control. Compared with 2011–2015, the level of the national industrial pollution mutualism degree index has deteriorated slightly from 2016 to 2020, and there is still room for improvement. The number of S-level provinces with industrial pollution rankings decreased slightly, while the number of E-level provinces increased. Ranked at the C-level, D-level and E-level, the pollution prevention and control speed of provinces with a low degree of mutualism did not match the speed of improvement in the socioeconomic development level. Provinces at this level of mutualism can improve by promoting the use of clean energy for production, upgrading packaging and products to use recyclable materials and other green production methods. According to different regions, the overall rank of the industrial pollution mutualism degree in the northeast and northern coastal areas was higher. The high rank of the northeast region may be the result of many years of pollution control, curbing environmental degradation, and sustainable development is an important goal of the strategy of revitalizing the old industrial base in the northeast. In recent years, the central government has attached great importance to the prevention and control of industrial pollution on the northern coast; investment in industrial pollution control occurs in a large proportion of the country and the emission intensity decreases significantly with the improvement of treatment capacity.

As shown in Figure 5, from 2016 to 2020, 8 provinces, including Tianjin and Nei Mongol, had an advanced degree of mutualism at the S-level, while 10 provinces, such as Hebei and Liaoning, had a low degree of mutualism at the E-level. This indicates that most provinces did not pay enough attention to the prevention and control of domestic pollution. Compared with 2011–2015, the number of provinces in the advanced mutualism degree with a domestic pollution rank of S-level from 2016 to 2020 decreased significantly, while the number of some provinces in the low and lower mutualism degree increased. The number of provinces at the D-level and E-level increased more, indicating that the mutualism between domestic pollution effectiveness and socioeconomic development in most provinces has deteriorated in recent years. From a regional point of view, most provinces in the middle reaches of the Yellow River had a higher rank for the prevention and control of domestic pollution. In recent years, the ecological protection of the Yellow River Basin has been the focus of national attention, and the effective treatment of domestic sewage in Nei Mongol, Henan and Shaanxi is the reason for its higher rank.

From 2016 to 2020, only Hubei, Shaanxi and Qinghai had an advanced S-level mutualism degree in both industrial and domestic pollution. Hebei, Liaoning, Jilin, Jiangxi, Xizang and Xinjiang had advanced and low mutualism degrees in industrial or domestic pollution from 2016 to 2020. Nei Mongol, Hunan and Sichuan had S-level and E-level domestic and industrial pollution in the target year, respectively, and these provinces had a large gap in the mutualism degree and rank of industrial and domestic pollution. From 2016 to 2020, most provinces were not at the S-level at the same time in two pollution aspects, and it was necessary to take into account the prevention and control of industrial and domestic pollution.

### 3.3. Spatial Autocorrelation and Spatial Heterogeneity Analysis of Ranks

According to the above rank analysis of industrial and domestic pollution, it can be seen that the rank gap between different regions in China is large. This study used the global and local Moran’s *I* for spatial autocorrelation analysis to discuss the spatial autocorrelation or spatial agglomeration characteristics of the rank. Both global and local Moran’s *I* uses *p*-value tests, the null hypothesis being that the sample is spatially randomly distributed, and the *p*-value was considered significant when it is less than 0.1. The global and local Moran’s *I* in this study were calculated using Geoda. The map was made using QGIS, and the *p*-value was obtained using 999 Monte Carlo permutations.

#### 3.3.1. The Space Weight Matrix Selection

When performing global or local spatial autocorrelation tests, it is first necessary to determine the space weight matrix. This is a symmetrical matrix with a principal diagonal of 0, which reflects the spatial connections between different samples. The rank data were polygon data, and the queen adjacency is commonly used in the polygon adjacency relationship to have a common boundary or node. The bidirectional matrix weight value of the two faces was taken as 1 when there was queen adjacency; otherwise, the weight value was 0. In this study, Geoda was used to establish a spatial weight matrix based on queen adjacency. There was no common boundary or node between Hainan Province and other regions in the study area, and according to the conventional method, the bidirectional weight value with Guangdong Province was set to 1 in the space weight matrix.

The differences in resource endowments, infrastructure, production factors, policies and systems in different regions, that is, differences in resource allocation, are important factors that lead to different ranks between regions. This study argues that if a province or municipality is adjacent to the administrative division of a province or municipality or is in the same area, the two have an adjacency relationship. Neighboring provinces will have a certain degree of similarity in resource endowment, factor allocation, etc., so they were divided according to the four economic regions announced by the National Bureau of Statistics. In this study, based on the space weight matrix established for each province according to the administrative division, different spatial weight matrices were set for each region. The bidirectional weight value of each province and the neighboring provinces was 1, and the weight value of the other provinces was 0. The following is based on the space weight matrix established by the four economic regions to calculate the global and local Moran’s *I* and analyze the spatial characteristics of industrial and domestic pollution ranks between provinces and regions.

#### 3.3.2. Global Spatial Autocorrelation Analysis

Global Moran’s *I* was proposed by the Australian statistician Moran in 1950 to measure the overall spatial autocorrelation properties in the study area. The average rank level of a province and its neighboring provinces was used to compare it with the national average rank level. Global Moran’s *I* reflected the average of the spatial autocorrelation degree of each province. When the global Moran’s *I* > 0, it means that the rank of the province and the neighboring province are relatively similar and there is a positive autocorrelation in the space representing agglomeration. When the global Moran’s *I* < 0, it means that the rank difference between the province and the neighboring province and city is high, and there is a negative spatial autocorrelation representing discreteness. To calculate the global Moran’s *I*, more than 30 samples are usually required to ensure the credibility of the index, and this study used 31 provinces as samples for analysis.

According to Table 4, the global Moran’s *I* of administrative divisions was analyzed first. The global Moran’s *I* of industrial and domestic pollution from 2011 to 2015 was significantly negative. This indicates that there was a global spatial negative autocorrelation between industrial and domestic pollution ranks, that is, the industrial and domestic pollution ranks of provinces with adjacent relationships in China had a large gap. From 2016 to 2020, the global Moran’s *I* of industrial and domestic pollution was not significant. This indicates that the ranks of industrial and domestic pollution in various provinces tended to be randomly distributed, and the regularity of the rank distribution was poor. In this study, we believe that a random distribution was better than a negative autocorrelation for the following reasons. First, from the perspective of spatial distribution, the discretization of spatial negative autocorrelation is a manifestation of uneven development, and random distribution indicates that the ranks of provinces are evenly distributed across the country. Second, from the perspective of the driving effect, the driving effect of high–high agglomeration is the strongest. The high–low agglomeration and low–high agglomeration of negative spatial autocorrelation represent the negative impact of a high-rated province on the province with adjacencies. A balanced random distribution indicates that there is no negative impact between adjacent provinces, so the random distribution follows the positive autocorrelation in terms of the driving effect, which is better than a negative autocorrelation.

The global Moran’s *I* under the division of the four economic regions was completely consistent with the positive and negative under the administrative division, and the difference in significance was large. This indicates that the differences in resource allocation between regions strengthened the spatial heterogeneity of industrial and domestic pollution ranks in different provinces to a certain extent. The global Moran’s *I* of domestic pollution ranks in the four economic regions changed significantly in different years in the opposite direction to administrative divisions. The spatial distribution of domestic pollution ranks changed from a random distribution in 2011–2015 to a significant spatial positive autocorrelation from 2016 to 2020. From the perspective of spatial heterogeneity, provinces in different regions have a driving effect on the coordination between domestic pollution and socioeconomic development of provinces with adjacent relationships.

#### 3.3.3. Local Spatial Autocorrelation Analysis

Global Moran’s *I* can test the spatial autocorrelation of the whole region from a macro perspective, but global spatial autocorrelation is not related to local spatial autocorrelation. This means that a certain agglomeration obtained by calculating global Moran’s *I* cannot indicate whether there is an agglomeration in every part of the whole region. In addition, whether the global Moran’s *I* is significant or not does not indicate whether local spatial autocorrelation exists from a meso perspective. There may be positive autocorrelation of some samples and negative autocorrelation of other samples in the whole region. The positive and negative autocorrelation cancellation makes the global Moran’s *I* value tend toward 0 and decrease the significance.

The local Moran’s *I*, also known as the Local Indicators of Spatial Association (Lisa), was proposed by Anselin in 1995. Compared with the global Moran’s *I*, the local Moran’s *I* can accurately determine the spatial autocorrelation of local samples, which is more statistically significant than the Moran scatterplot. In the output clustering category, a high–high agglomeration indicates that the attribute values of the current provinces and related regions are higher than the attribute values of the whole region. A high–low agglomeration indicates that the attribute value for that region is higher than the average attribute value for the region to which it is related.

Table 5 shows significant local Moran’s *I p*-values for all provinces. The local Moran’s *I* of most provinces from 2011 to 2020 showed a negative spatial autocorrelation in terms of industrial pollution rank. This indicates that these provinces had higher socioeconomic development and industrial pollution ranks, while neighboring provinces had lower ranks, which further verifies the results of the global Moran’s *I* above. In terms of domestic pollution rank, the local Moran’s *I* of all provinces from 2011 to 2015 showed a negative spatial autocorrelation. The agglomeration phenomenon in Chongqing, Gansu and Shaanxi from 2016 to 2020 showed that Gansu had a higher rank during this period and a lower rank in neighboring provinces, and Gansu was a typical area with a high rank. In terms of industrial and domestic pollution, there was no significant high–high agglomeration in provinces, and most provinces were negatively correlated in space. Each province and city with a higher rank should drive the development of surrounding areas to a high–high agglomeration state and promote the coordination between socioeconomic development and pollution prevention and control in neighboring provinces.

According to Table 6, the spatial autocorrelation results corresponding to the local Moran’s *I* under the four economic regions were quite different from those of administrative divisions. Under the division of the four economic regions, some provinces on the eastern coast were in a state of high–high agglomeration in terms of domestic pollution rank from 2016 to 2020, while high–low agglomerations were widespread in the western region. There was significant spatial heterogeneity in the target year of domestic pollution rank. All high-rated provinces in the western region should encourage neighboring provinces to explore new models of linkage development while coordinating their own development. High-rated provinces in the eastern region should expand their current driving role and promote the improvement of industrial and domestic pollution ranks in the region and even the whole country.

## 4. Conclusions and Policy Recommendations

### 4.1. Conclusions

The conclusions of this study are as follows:(1)The proportion of HDI and the overall change trend of industrial and domestic pollution effectiveness indicators in the country from 2011 to 2020 were similar, and the proportion or effectiveness index value of approximately half of the provinces increased. The proportion of HDI in most provinces in the target year range did not change substantially, indicating that the socioeconomic development rate of all provinces remained relatively consistent. The indicators of industrial and domestic pollution effectiveness in some areas with high-level socioeconomic development showed an upward trend, that is, pollution emissions decreased. However, the indicators of industrial and domestic pollution efficacy in some areas with rapid socioeconomic development showed a downward trend.(2)The degree of coordination between industrial pollution prevention and control and socioeconomic development in various provinces was obviously better than that of domestic pollution. Compared with 2011–2015, the number of provinces with mutual promotion of the HDI and industrial pollution prevention and control effectiveness decreased slightly from 2016 to 2020, while the number of provinces with HDI and domestic pollution prevention and control effectiveness decreased significantly. The overall rate of industrial pollution prevention and control in the country was slightly unbalanced compared with the growth rate of socioeconomic development. The speed of domestic pollution prevention and control did not match the growth rate of socioeconomic development.(3)Except for Hubei, Shaanxi and Qinghai, which had an advanced degree of mutualism of industry and domestic pollution at the same time, most provinces were not at the same S-level in terms of their industrial and domestic pollution ranks. From 2016 to 2020, 14 provinces were in the advanced mutualism degree of pollution and socioeconomic development, that is, at the S-level. Eleven provinces had industrial pollution ranks at the D-level or E-level, and eight provinces had a low degree of mutualism. In terms of the mutualism degree index and ranks of domestic pollution, from 2016 to 2020, there were eight provinces rated S-level in the advanced mutualism degree of pollution and socioeconomic development. Fifteen provinces had a pollution rank at the D-level or E-level, and ten provinces had a low degree of mutualism.(4)The spatial distribution of ranks tended to be generally balanced. Most provinces with significant agglomerations had a negative impact on neighboring provinces. There was significant spatial heterogeneity between the four economic regions. From 2016 to 2020, the global Moran’s *I* of industrial and domestic pollution ranks based on administrative division was randomly distributed, and there were no high–high type provinces with a leading role in the local Moran’s *I*. The local Moran’s *I* of domestic pollution rank under the four economic regions in the target year showed that some eastern coastal provinces had a positive spatial autocorrelation of the high–high type, while the provinces in the western region mainly had a negative spatial autocorrelation of the high–low type.

### 4.2. Policy Recommendations

Improving the quality of the ecological environment is an important prerequisite for promoting high-quality development in the new era. The deterioration of the ecological environment can be said to have been caused by excessive discharge of pollutants in the process of socioeconomic development. The coordinated control of multiple pollutants and the regional coordinated treatment of pollution is an important way to improve the quality of the ecological environment. This study analyzed industrial and domestic pollution as a breakthrough in the ecological environment, and the conclusions drawn in the coordination rank and spatial distribution of socioeconomic development and industrial and domestic pollution prevention and control can promote green and high-quality development. Based on the previous analysis, the following policy recommendations are made:(1)Provinces can adjust their socioeconomic development and pollution prevention policies according to their ranks so that the two can develop at a higher level of coordination. S-level provinces should continue to maintain the current level of coordination of socioeconomic development and pollution prevention and control. A-level and C-level provinces should coordinate the speed of industrial and domestic pollution prevention and control based on the level of socioeconomic development. Compared with A-level provinces, C-level provinces should pay more attention to the coordination of the two through the effective combination of the role of the government and the market, rationally allocate all kinds of public and social resources and vigorously promote socioeconomic development, while maintaining the existing speed of pollution prevention and control. B-level and D-level provinces should strengthen the prevention and control of industrial and domestic pollution, and D-level provinces should increase pollution prevention and control. In terms of industrial pollution, provinces at these two levels can promote the use of new technologies and new environmentally friendly energy sources, optimize the industrial structure and promote industrial transformation and upgrades to prevent and control pollution. In terms of domestic pollution, the government can improve the public transportation network, advocate people to develop a green and low-carbon lifestyle and promote the harmlessness and recycling of garbage to reduce pollution emissions in all aspects. E-level provinces should reasonably adopt the methods mentioned above to improve the coordination level of socioeconomic development and pollution prevention and control. These provinces can first adopt similar measures to B-level and D-level provinces to prevent and control pollution, and then coordinate the speed of industrial and domestic pollution prevention and control according to the level of socioeconomic development.(2)The government should pay attention to the prevention and control of industrial and domestic pollution at the same frequency to achieve the effect of improving both. It is necessary to reasonably formulate domestic pollution control plans and implementation plans, improve management systems and mechanisms and improve the emission standard and normative system. In terms of investment in pollution prevention and control, government departments should refine the amount of investment in domestic pollution. Some major polluting provinces should appropriately increase the amount of investment and simultaneously strengthen the enforcement of illegal domestic pollution. According to the rank results, it can be seen that different provinces have different emphases and effects on the prevention and control of industrial and domestic pollution. There is no domestic pollution source index in the amount of current pollution investment; thus, the statistical department should refine the domestic pollution prevention and control investment index. In addition, the amount of investment in industrial pollution prevention and control varies greatly between provinces and across different years within the same province and is not obviously related to pollution emissions. Major polluting provinces should adjust the amount of annual investment according to the amount of pollution discharge.(3)Combined with the spatial nature of the ranking, targeted regional linkage policies should be formulated from the perspective of the whole. The eastern and western regions should appropriately tilt resources from high-rank provinces to provinces negatively related to the surrounding space in the point axis mode to promote the coordinated progress of pollution prevention and social development in their neighboring regions to improve the overall development strength and coordination level of the region. The difference in resource allocation between the four economic regions is an important reason for the significant regional heterogeneity in the eastern region. The eastern region should improve the coordination mechanism of socioeconomic development and pollution prevention and control in terms of domestic pollution, amplify regional advantages, continue to deepen the linkage and cooperation with the central and western regions and realize the full docking of their respective advantages and the smooth flow of resources and elements. High-rank provinces in the central and western regions that have not played a leading role in industrial and domestic pollution should implement the strategy of regional cooperation in pollution prevention and control. Provinces with lower ranks should rely on the advantages of the policy tilt, strengthen intraregional linkages and exchange pollution prevention and control experience with high-rated provinces to improve the coordination level of their own socioeconomic development and industrial and domestic pollution prevention and control.

## Figures and Tables

**Figure 1 ijerph-20-04561-f001:**
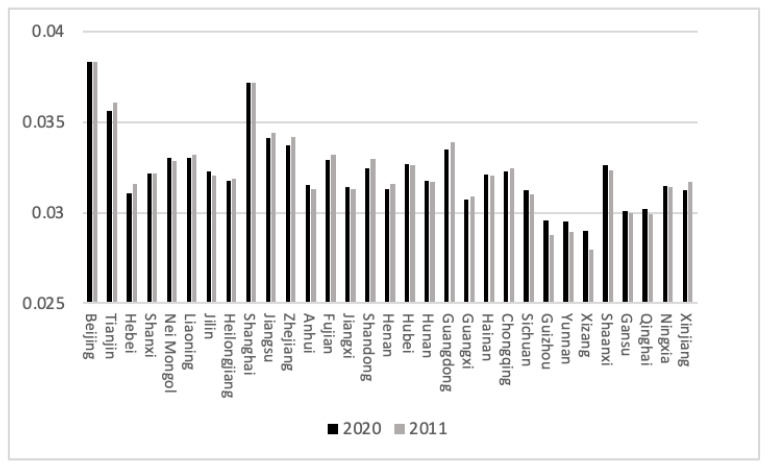
The proportion of HDI in 2011 and 2020. (Source: made by the authors).

**Figure 2 ijerph-20-04561-f002:**
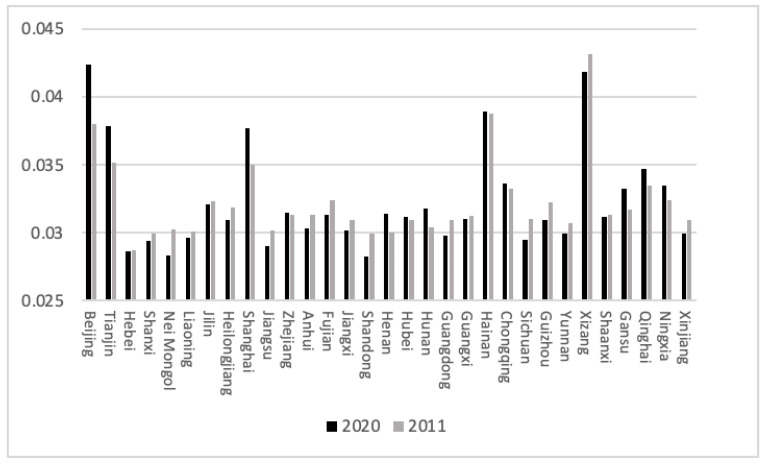
The effectiveness indicators of industrial pollution. (Source: made by the authors).

**Figure 3 ijerph-20-04561-f003:**
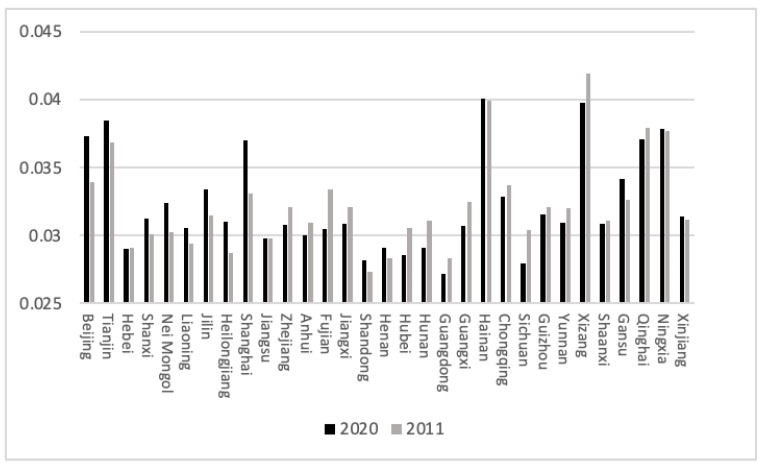
The effectiveness indicators of domestic pollution. (Source: made by the authors).

**Figure 4 ijerph-20-04561-f004:**
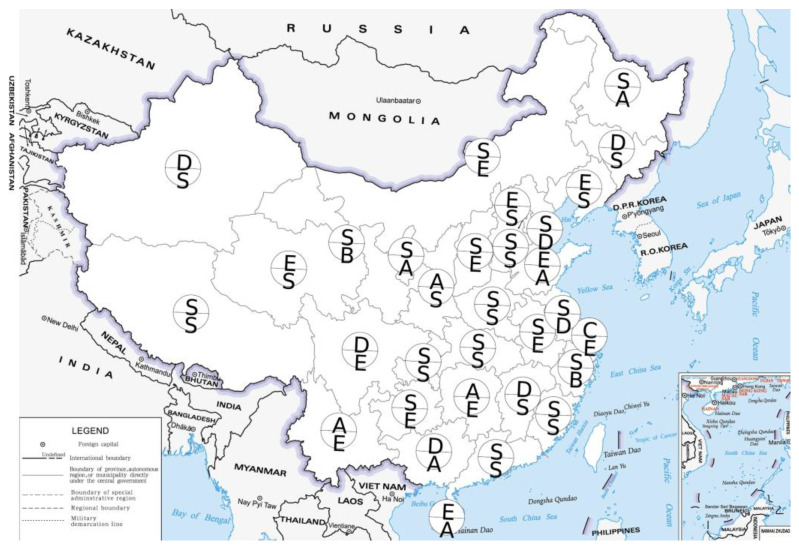
Ranks of industrial pollution effectiveness. (Note: the map is made based on the GS (2019) 1679, available online: http://bzdt.ch.mnr.gov.cn, accessed on 17 February 2023).

**Figure 5 ijerph-20-04561-f005:**
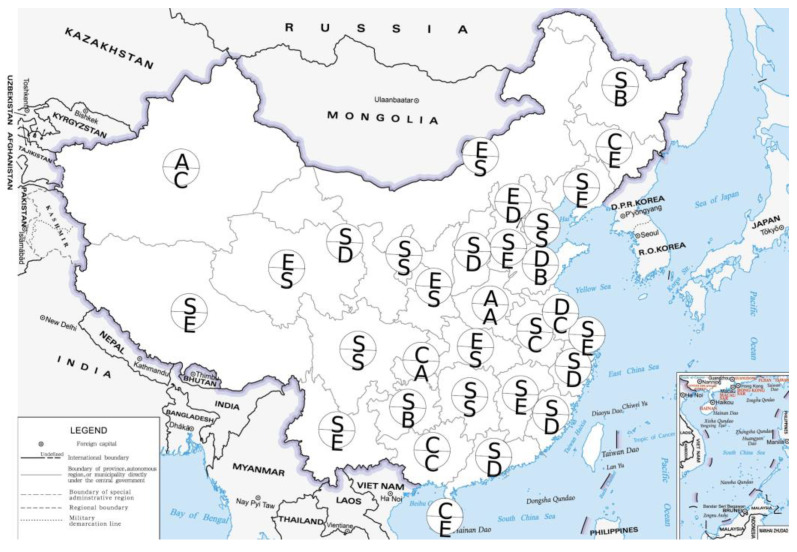
Ranks of domestic pollution effectiveness. (Note: the map was made based on the GS (2019) 1679, available online: http://bzdt.ch.mnr.gov.cn, accessed on 17 February 2023).

**Table 1 ijerph-20-04561-t001:** HDI and industrial pollution effectiveness ranking basis.

Mutualism Degree	Force-On Indices	Mutualism Degree Index	Ranks
Advanced mutualism degree	S_I_(t) > 0, S_H_(t) > 0	S_1_(t) > 1	S-level
Higher mutualism degree	S_I_(t) > 0, S_H_(t) < 0, |S_H_(t)| < S_I_(t)	0 < S_1_(t) < 1	A-level
	S_I_(t) < 0, S_H_(t) > 0, |S_I_(t)| < S_H_(t)	0 < S_1_(t) < 1	B-level
Lower mutualism degree	S_I_(t) > 0, S_H_(t) < 0, S_I_(t)< |S_H_(t)|	−1 < S_1_(t) < 0	C-level
	S_I_(t) < 0, S_H_(t) > 0, S_H_(t) < |S_I_(t)|	−1 < S_1_(t) < 0	D-level
Low mutualism degree	S_I_(t) < 0, S_H_(t) < 0	S_1_(t) < −1	E-level

Note: referring to the research of Chen et al. [34] and related studies, we set the ranks based on the research needs.

**Table 2 ijerph-20-04561-t002:** Classification of the force-on index of industrial pollution effectiveness.

Classification Basis	Explanation of the Classification	2011–2015	2016–2020
S_I_(t) > 0, S_H_(t) > 0	Industrial pollution effectiveness and HDI reinforce each other	Tianjin, Hebei, Shanxi, Nei Mongol, Heilongjiang, Jiangsu, Zhejiang, Anhui, Fujian, Henan, Hubei, Guangdong, Chongqing, Guizhou, Xizang, Gansu, Ningxia	Beijing, Hebei, Liaoning, Jilin, Fujian, Jiangxi, Henan, Hubei, Guangdong, Chongqing, Xizang, Shaanxi, Qinghai, Xinjiang
S_I_(t) > 0, S_H_(t) < 0	HDI reinforces industrial pollution effectiveness while industrial pollution effectiveness suppresses HDI	Shanghai, Hunan, Yunnan, Shaanxi	Heilongjiang, Shandong, Guangxi, Hainan, Ningxia
S_I_(t) < 0, S_H_(t) > 0	HDI suppresses industrial pollution effectiveness while industrial pollution effectiveness reinforces HDI	Jilin, Jiangxi, Guangxi, Sichuan, Xinjiang	Tianjin, Jiangsu, Zhejiang, Gansu
S_I_(t) < 0, S_H_(t) < 0	Industrial pollution effectiveness and HDI suppress each other	Beijing, Liaoning, Shandong, Hainan, Qinghai	Shanxi, Nei Mongol, Shanghai, Anhui, Hunan, Sichuan, Guizhou, Yunnan

Source: made by the authors.

**Table 3 ijerph-20-04561-t003:** Classification of the force-on index of domestic pollution effectiveness.

Classification Basis	Explanation of the Classification	2011–2015	2016–2020
S_D_(t) > 0, S_L_(t) > 0	Domestic pollution effectiveness and HDI reinforce each other	Tianjin, Hebei, Shanxi, Liaoning, Heilongjiang, Shanghai, Zhejiang, Anhui, Fujian, Jiangxi, Hunan, Guangdong, Sichuan, Guizhou, Yunnan, Xizang, Gansu, Ningxia	Tianjin, Nei Mongol, Hubei, Hunan, Sichuan, Shaanxi, Qinghai, Ningxia
S_D_(t) > 0, S_L_(t) < 0	HDI reinforces domestic pollution effectiveness while domestic pollution effectiveness suppresses HDI	Jilin, Henan, Guangxi, Hainan, Chongqing, Xinjiang	Jiangsu, Anhui, Henan, Guangxi, Chongqing, Xinjiang
S_D_(t) < 0, S_L_(t) > 0	HDI suppresses domestic pollution effectiveness while domestic pollution effectiveness reinforces HDI	Jiangsu, Shandong	Beijing, Shanxi, Heilongjiang, Zhejiang, Fujian, Shandong, Guangdong, Guizhou, Gansu
S_D_(t) < 0, S_L_(t) < 0	Domestic pollution effectiveness and HDI suppress each other	Beijing, Nei Mongol, Hubei, Shaanxi, Qinghai	Hebei, Liaoning, Jilin, Shanghai, Jiangxi, Hainan, Yunnan, Xizang

Source: made by the authors.

**Table 4 ijerph-20-04561-t004:** Global Moran’s *I* of the socioeconomic development and pollution level.

	Rank of Industrial Pollution in 2011–2015	Rank of Industrial Pollution in 2016–2020	Rank of Domestic Pollution in 2011–2015	Rank of Domestic Pollution in 2016–2020
Moran’s *I* of administrative division	−0.199	−0.001	−0.222	0.104
*p*-value of administrative division	0.06	0.39	0.04	0.15
Moran’s *I* of four economic regions	−0.103	−0.054	−0.09	0.068
*p*-value of four economic regions	0.13	0.46	0.21	0.09

Source: made by the authors.

**Table 5 ijerph-20-04561-t005:** Local Moran’s *I* of ranks in administrative division.

Types	Rank of Industrial Pollution in 2011–2015	Rank of Industrial Pollution in 2016–2020	Rank of Domestic Pollution in 2011–2015	Rank of Domestic Pollution in 2016–2020
High–high	None	None	None	None
Low–low	Shaanxi	None	None	Chongqing, Shaanxi
Low–high	Xizang	Guangxi	Heilongjiang, Ningxia	Tianjin
High–low	Beijing, Shanghai, Jiangxi, Shandong, Hainan	Tianjin	Beijing, Guangxi, Hainan	Gansu

Source: made by the authors.

**Table 6 ijerph-20-04561-t006:** Local Moran’s *I* of ranks in four economic regions.

Types	Rank of Industrial Pollution in 2011–2015	Rank of Industrial Pollution in 2016–2020	Rank of Domestic Pollution in 2011–2015	Rank of Domestic Pollution in 2016–2020
High–high	None	None	None	Jiangsu, Zhejiang, Fujian
Low–low	None	None	None	Guizhou, Chongqing, Shaanxi
Low–high	None	None	None	Tianjin
High–low	Jiangxi	None	None	Guangxi, Yunnan, Xizang, Gansu, Xinjiang

Source: made by the authors.

## Data Availability

The data were obtained from the China Statistical Yearbook (http://www.stats.gov.cn/tjsj./ndsj/ accessed on 17 February 2023) and China Statistical Yearbook on Environment (https://navi.cnki.net/knavi/yearbooks/YHJSD/detail?uniplatform=NZKPT accessed on 17 February 2023).

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
