# Peer review of "Research on the Evaluation and Spatial Characteristics of China’s Provincial Socioeconomic Development and Pollution Control Based on the Lotka–Volterra Model"

_ijerph, 2023, doi:10.3390/ijerph20054561_

Round 1

Reviewer 1 Report

The study used the Lotka-Volterra Model to estimate regional socioeconomic development in 31 Chinese provinces. It also examined industrial and domestic pollution as an ecological breakthrough. The study derives its findings from the coordination rank and spatial distribution of socioeconomic development and industrial and domestic pollution prevention. Based on their analysis, the authors also make policy recommendations. 

I have some major comments for the author's consideration.

1. Sentences in the abstract are too long. That might cause difficulties for the readers. I strongly recommend rephrasing the abstract to make it clearer and more precise. 

One important point to note is that several sentences are too long throughout the text. Authors need to consider splitting those sentences. 

2. Line 37-40: Reference is missing. It could be a report or a web link available in the public domain so other scientists can refer to it for dataset/information.

3. Line 41-45: Reference missing.

4. Line 51-52: "CO2" -> "CO2"  and "NOx" -> "NOx" (Kindly make these changes throughout the text).

5. Line 55-59: Needs further explanation 

6. Line 59: "PM2.5" -> "PM2.5"

7. Line 85-87: Needs elaboration, especially for the readers.

8. A brief introduction to HDI will be helpful for the readers.

9. Just a suggestion, a summary of the following sections can be provided as the last paragraph of the introduction.

10. Line 128-129: 2011-2020 China Statistical Yearbook, is it open-source information? Could you please provide a reference or a web link for this source?

11. It is highly recommended that authors provide a list of acronyms used in the manuscript just before the reference section.

12. Line 181-184: Any specific reasons why volatile organic compounds (VOCs) are not considered? 

Reviewer 2 Report

No remarks to the authors. The authors of the study undertook a grandiose research for their country. The list of reference is too long, to my opinion. there are only Chinese name there. Does it mean that similar studies were not performed in other countries? In this case the authors should emphasied their pioneer role in accessement of pollution in provincial and not only regions. I recommend to publish this paper as it is.

I examined the websites of the Authors and find them competent for the conclusions and recommendations they expressed in their paper.

The first author, Xue Zhou, is a young lady who recently defended her PhD, and the corresponding authors, Prof. Jiapeng Wang , seems her supervisor. The list of their publications shows their competence in the subject of the given paper. 

China is making great efforts to improve their stressful environmental situation, and it is reflected in funding such studies as this one. This very research was funded by the Ministry of Education in China (Humanities and Social Sciences), I don't exclude that this paper is a kind of a Project Report of the authors. In support of their activity, I suggest to publish this paper as it is. It's a huge research, buy the way!

Reviewer 3 Report

Title: Research on the Evaluation and Spatial Characteristics of Provincial Socioeconomic Development and Pollution Control  Based on the Lotka-Volterra Model

 Xue Zhou, Jiapeng Wang

The study analyse  degree of mutualism between socioeconomic development and industrial  and domestic pollution in provinces of China and the spatial characteristics.

Abstract: Lines 8-22

Please structure the Abstract as:

Introduction-Aims

Method

Results and interpretation

Introduction

The aim of the study is not very clear expressed. Please revise

Literature review should be updated to 2023 and must be internationalized. Please do a analytical critical review of international and from China related with your topic.

Please analyses critically the findings of the articles and the limitations.

Please indicate also at least tree similar article to your research published recently (last 5 years).

Eq no 1 line 144: please insert the datasource and explain what represent each component

Figure 1 line 157: Please insert the datasource. Insert the units for vertical axis. Update to 2023.

Fig.2, 3 idem as fig 1

The models for industrial pollution is generally expressed as follows: line 274 - Please insert the datasource

Eq 5-12: insert the authors

Table 1 line 303: insert the datasource

Table 2 line 316: update to 2023 and insert the datasource

Table no 3, line 332: idem as above

 Formula no 2, line 190: please insert the datasource.

Fig. 4 line 379:

Please insert some toponyms, the neighbors

Missing the legend

Insert the datasource

Fig.5 line 403- idem as above

Tables 5 and 5 please update to 2023 and insert the datasource

Conclusions: should report the data updated to 2023. Please revise.

References: update the references to 2023

Please cite also: 10.30638/eemj.2018.272

Is it relevant and
interesting?

The paper is relevant for postpandemic period. It will synthetize better the actual available international literature data, focus mainly on pollution and high-efficient development and industrial structure is a key factor in this dynamic environment of China.

How original is the topic?

Is an actual topic even not high level of originality; but deal with an important subject in the post pandemic period

What does it add to the subject
area compared with other published material?

The paper should be better documented (number of only 28 scientific published articles in references list), very few updated to 2023.

Is the paper well written?

The paper is well written. The quality of English translation is good.

Is the text clear and easy to read?

The text is well structured, clear and easy to read from the specialists in the field but as well as from the persons from public.

Are the conclusions consistent with the evidence and arguments presented?

The conclusions are well represented in the paper. The authors underline the degree of mutualism between socioeconomic development and industrial and domestic pollution in provinces of China, in conditions of influencing factors in different environment milieu, important  especially for human communities.

Best regards,

February 2023

Reviewer 4 Report

The manuscript in a very interesting way combined the problems of the socio-economic type with the natural phenomenon through the application of the population model. Very imaginative work. A bold approach to the interpretation of results that is worth presenting to a wider scientific audience.

Round 2

Reviewer 1 Report

Thank you for addressing most of the comments in the review report.